# Broadband Multidimensional Spectroscopy Identifies the Amide II Vibrations in Silkworm Films

**DOI:** 10.3390/molecules27196275

**Published:** 2022-09-23

**Authors:** Adam S. Chatterley, Peter Laity, Chris Holland, Tobias Weidner, Sander Woutersen, Giulia Giubertoni

**Affiliations:** 1Department of Chemistry, Aarhus University, 8000 Aarhus C, Denmark; 2Department of Materials Science and Engineering, University of Sheffield, Sheffield S1 3JD, UK; 3Van ‘t Hoff Institute for Molecular Sciences, University of Amsterdam, 1098 XH Amsterdam, The Netherlands

**Keywords:** (2D)-infrared spectroscopy, amide II, secondary structure, *Bombyx mori* native silk films

## Abstract

We used two-dimensional infrared spectroscopy to disentangle the broad infrared band in the amide II vibrational regions of *Bombyx mori* native silk films, identifying the single amide II modes and correlating them to specific secondary structure. Amide I and amide II modes have a strong vibrational coupling, which manifests as cross-peaks in 2D infrared spectra with frequencies determined by both the amide I and amide II frequencies of the same secondary structure. By cross referencing with well-known amide I assignments, we determined that the amide II (N-H) absorbs at around 1552 and at 1530 cm^–1^ for helical and β-sheet structures, respectively. We also observed a peak at 1517 cm^−1^ that could not be easily assigned to an amide II mode, and instead we tentatively assigned it to a Tyrosine sidechain. These results stand in contrast with previous findings from linear infrared spectroscopy, highlighting the ability of multidimensional spectroscopy for untangling convoluted spectra, and suggesting the need for caution when assigning silk amide II spectra.

## 1. Introduction

The use of linear infrared spectroscopy to study the molecular structure of protein in unpurified biological materials like silks is very common [1,2,3,4]. Among all vibrations, the most studied is the amide I vibration, which consists mostly of the stretching of the carbonyl. Amide I vibrations strongly absorb between 1600–1700 cm^–1^ and they are particular sensitive to the secondary structure adopted by the protein [5,6]. For instance, proteins adopting a β-sheet structure strongly absorb infrared light at 1620 cm^–1^, while those adopting an α-helix conformation show a strong absorption band at 1660 cm^–1^ [7]. Although researchers mostly use amide I vibrations to investigate the secondary structures of proteins, amide groups have additional vibrations, such as the amide II, that can be also used to investigate molecular properties. The amide II band absorption is due to a combination of the C-N stretching and N-H bending motions, and it generally absorbs at around 1550 cm^–1^ [5,6,8]. 

Similar to amide I, the absorption frequency of the amide II mode depends on the adopted secondary structure [8]. One of the advantages of the amide II region is that the vibrational frequency is strongly affected by isotopic exchange, which can be induced by the use of D_2_O instead of water. Indeed, by replacing N-H with N-D, the amide II frequency decreases by ~100 cm^−1^, producing the so-called amide II’ bands [6,9]. The strong isotope effect of amide II modes can be used to investigate solvent effects on the molecular properties of biomaterials, which is a highly important research question especially with silk-based materials, such as films cast from the gland contents of the silkworm [7]. Such *Bombyx mori* native silk fibroin films are composed of mixture of non-crystalline (amorphous) and crystalline regions, which are expected to have different solvent accessibility because of the different protein packing. β-sheet appears to be the most stable form and dominates the crystalline content of natural silk fibers [10], while helical and random structures can be observed in the non-crystalline material [11]. Thanks to the strong isotope effect of the amide II vibration, isotope-exchange experiments can help to understand the effect of the exposure to solvent, i.e., to water, on the protein structures, and to determine the degree of solvent accessibility in the different regions [2]. Unfortunately, these experiments require knowledge of the exact vibrational frequencies of the amide II modes of the different secondary structures. In case of *B. mori* silk, the assignment of amide II modes to specific conformations is not unique in literature because the amide II vibrational band is more congested than the amide I band [2,12,13,14,15,16,17,18,19].

To solve this, we can use two-dimensional infrared spectroscopy (2DIR) [20,21,22,23,24]. Compared to linear infrared where the absorption of the infrared beam is recorded, 2DIR spectroscopy is nonlinear spectroscopy that reveals the change in the infrared spectrum as a function of the frequency of a preceding infrared pump laser pulse. It is the infrared analogue of 2D-NMR. By using a broadband pump pulse, we simultaneously excite a range of frequencies, and the obtained 2D spectra show the pump induced absorption change ∆A(*ω_pump_*, *ω_probe_*) as a function of the pump and probe frequencies. Our 2DIR spectrometer operates in the time-domain, but is converted into the frequency domain by Fourier transform. For the purposes of this work, we need only consider the frequency domain. 

We recently used 2DIR to investigate the secondary structure of silk [7]. We found evidence that at ambient humidity, silk protein contains helical, β-sheet and random coil structures, with helical being the predominant structure (>70%). We found that helical structure absorbs at 1660 cm^–1^, while β-sheet absorbs at 1625 cm^–1^ and 1695 cm^–1^, and random coil at 1640 cm^–1^. The precise assignment of these vibrational bands was made possible by selecting specific polarization combinations in 2DIR spectroscopy where we can obtain unique spectral signatures that allow us to disentangle and assign vibrational bands to specific secondary structures. Unfortunately, amide II bands do not show similar spectral signatures that allow us to correlate in an unambiguous manner to specific secondary structures. However, amide II and amide I are strongly coupled as they belong to the same functional group [25,26], and upon excitation of one, the vibrations of the second are strongly affected, leading to the appearance of characteristic off-diagonal cross-peak features (Figure 1). The magnitude of the coupling between these two modes will be strongest between amide I and amide II modes located in the same secondary structure. Hence, using the amide II/amide I cross peaks it becomes possible to relate amide II bands to specific amide I bands, and thus to assign them to specific secondary structures [25,26].

Here, we apply this broadband 2DIR approach to investigate in situ the amide II vibrations present in films produced using unpurified protein from by *B. mori* caterpillars. By exciting amide I (or amide II) and probing at amide II (or amide I), downward (or upward) cross-peaks appear due to vibrational coupling between amide I and amide II. Because of the strong coupling, the cross-peaks appear between amide modes belonging to the same secondary structure, and using known amide I assignments, we disentangle and assign amide II vibrational bands to either helical or β-sheet structures. 

## 2. Results

We begin with a brief overview of how to interpret a 2DIR spectrum [22]. Figure 1 shows a sketch of an ideal 2DIR spectrum consisting of two coupled vibrational modes labelled A and B. When the pump frequency is resonant with the ν_A_ = 1 ← ν_A_ = 0 transition, a fraction of molecules are excited to the ν_A_ = 1 state of this mode, resulting in a decrease in absorption (bleaching) at the ν_A_ = 1 ← ν_A_ = 0 frequency (Figure 1a). This gives rise to a negative (blue) bleach peak on the diagonal, with both pump and probe frequencies equal to A’s fundamental frequency. As the first vibrational level is now populated, a new absorption appears at the ν_A_ = 2 ← ν_A_ = 1 frequency, giving rise to positive (red) excited state absorption features. These peaks always appear at lower frequency than the bleach, due to anharmonicity. A similar diagonal pair appears when one pumps and probes mode B. However, if after excitation to ν_A_ = 1, we probe at the fundamental frequency of the B mode, anharmonic coupling will shift the ν_B_ = 1 ← ν_B_ = 0 transition to a lower frequency. This creates an off-diagonal pair of positive and negative features, where the positive features shows the new absorption frequency, and the negative feature shows the bleach where the absorption would have been if mode A had not been excited. The pump frequency of these cross-peaks corresponds to the ν_A_ = 1 ← ν_A_ = 0 transition, while the probe frequencies show the ν_B_ = 1 ← ν_B_ = 0 transition, under the influence of the anharmonic coupling. If the first mode excited is higher in frequency we refer to this pair as a *downward cross-peak*, and if it is lower we refer to it as an *upwards cross-peak*. Cross-peaks are only present between two modes if they are vibrationally coupled, which in practical terms means the two vibrational modes must be mechanically coupled, hence we can use them to assign secondary structure of unknown peaks if we have already determined the structure for the partner in a cross-peak [20,21,22,23,24].

Here, the ‘A’ and ‘B’ modes are the amide I and amide II bands of the silk-protein backbone, shown in Figure 1c. As mentioned above, the amide I vibrations (at around 1600–1700 cm^−1^) reflect mostly the C=O stretch of the peptide bond. The amide II vibration is at around 1500–1600 cm^−1^ and has a mixture of mostly N-H bend with some C-N stretch character. When the amide is deuterated, because of the mass change, the amide II mode localized more on the peptide bond, and the frequency is redshifted by around 100 cm^−1^. The amide I and II transition dipoles (shown as arrows in Figure 1c) lie at an angle of around 40° (70° for the amide II’) to each other [27].

The 2DIR spectrum of the silk film, after 36 h in a humid D_2_O environment, is shown in Figure 2, along with the linear FTIR spectrum. The pump and probe beams were polarized perpendicular to one another. To reduce spectral congestion, the silk films were partially deuterium exchanged, which removed some of the intensity from the amide II region (with a corresponding increase in the amide II’ region), which makes assignment of the amide peaks significantly easier. In the Appendix A, we show a 2DIR spectrum before deuteration; the amide II cross-peaks are much harder to resolve. The relative FTIR peak heights in the amide II and II’ regions show that the 36-h treatment was sufficient to exchange approximately half the accessible H atoms into D (see Appendix A). As mentioned above, peaks in 2DIR spectra are always present as a doublet of a positive (red) peak at a lower probe wavelength than a negative (blue) peak. The negative peak represents depletion of absorption after modes at that frequency have been excited, while the positive peak shows the new absorption frequency, which is generally redshifted due to anharmonicity. In the reported 2DIR spectrum, three vibrational regions are clearly visible: the amide II’ region around 1400–1500 cm^−1^, the amide II region around 1500–1600 cm^−1^, and the amide I region around 1600–1700 cm^−1^. These three regions combine to give nine sectors in a 2D spectrum. The sectors on the diagonal show the response pumping and probing the same vibrations; in this case, we clearly see three diagonal regions where we pump and probe separately the amide II’, amide II and amide I vibrations (from bottom left to top right). The off-diagonal sectors, which we will refer to as cross-peak regions, show the vibrational response after excitation in a different region, e.g., the top-left sector (I/II’) reports the change in amide II’ modes after the amide I region is excited. We divide these regions into upward and downward cross-peaks regions, where upward cross-peaks indicate that the excited vibrations is at a lower frequency with respect to the probed ones, and downward cross-peaks vice versa. When referring to a location on the 2D spectrum, it will always be in the format $\left(\omega_{pump},\;\omega_{probe}\right)$.

The diagonal amide I sector (I/I, top right) was the subject of our previous paper, so we will only recap the findings in brief [7]. An intense peak centered around 1650 to 1660 cm^−1^ in protein spectra has been assigned previously to α-helical modes [5,6,9,28,29] although more recent work on silk suggests that other helical structures (e.g., type II β-turns) may be involved [11,30,31] The peaks at 1620 cm^−1^ and 1695 cm^−1^ are the highly split amide I modes of a β-sheet, with off-diagonal peaks at (1620, 1695) and (1695, 1620) cm^−1^ showing the coupling between them. Buried in the congested diagonal spectrum is a peak at 1640 cm^−1^, which stems from the remaining randomly coiled structures.

We now move to the diagonal amide II and amide II’ sectors (sector (II/II) and (II’/II’), respectively). In order to see the diagonals more clearly, we plot a cut along the diagonal peak for these two sectors, together with the FTIR spectra. When compared to the linear FTIR spectrum, we can already appreciate that the spectral signatures in the corresponding 2DIR regions are less congested and better resolved. This is because the diagonal spectrum scales as µ^4^, as opposed to the FTIR spectrum, which scales as µ^2^, where µ is the dipole moment of the vibrational transition. Note that the presence of excited state absorptions can slightly shift the apparent peak centers found by diagonal cuts, but the general positions will be trustworthy [32]. In the amide II spectrum (Figure 3a), the diagonal slice shows much sharper peaks than the FTIR spectrum, allowing us to clearly resolve three distinct peaks: a strong sharp peak at 1517 cm^−1^, a weak feature at 1532 cm^−1^, and a strong asymmetric peak at 1556 cm^−1^. The amide II’ 2DIR diagonal cut spectra (Figure 3b) looks substantially different from the amide II. The 2D diagonal spectrum appears to be narrower than the FTIR, showing a main peak at 1475 cm^−1^ with a shoulder around 1455 cm^−1^. The absence of features below 1450 cm^−1^ in the 2D spectrum is due an experimental artifact stemming from the limited bandwidth in the 2DIR experiment: the red edge of the excitation laser pulse drops off rapidly in intensity below ~1480 cm^−1^ (see Appendix A). The intensity of a 2DIR spectrum is proportional to the intensity of the pump pulse [22], so the spectrum in the region below 1480 cm^−1^ is significantly reduced by the weak laser power. Despite this, between the FTIR and 2DIR diagonal spectra we can identify features centred at around 1410, 1440, 1450, 1460 and 1475 cm^−1^, although the precise locations are harder to identify in this case. By comparing the FTIR spectrum of the dry and treated samples (see Appendix A), we find that peaks at 1410 and 1450 cm^−1^ are already present prior to deuteration, indicating that these two bands are due to sidechain modes. Due to the limited bandwidth, little information is available from the amide II’ region, however we include the analysis for completeness in the Appendix A.

Using the diagonal cuts in the diagonal sectors, we were able to disentangle the congested FTIR spectrum and found that we have three modes in the amide II region absorbing at 1517, 1532 and 1556 cm^−1^, and one intense mode in the amide II’ region absorbing at 1475 cm^−1^. Although these modes absorb at distinct frequencies, their absorption frequencies are not enough to assign them to specific secondary structures. In order to do these, we exploit the fact that amide I and amide II modes are strongly coupled, and as previously discussed, cross-peaks are more intense between amide modes belonging to the same secondary structure.

By analyzing the cross peaks between the amide II bands and the amide I bands, and using the well-established assignments of the amide I bands [5,6], we thus can assign the features located in the amide II region. Figure 4 shows zooms of the two amide I/II cross peak regions (see Figure 4a,b), along with cuts through these spectra intersecting either the helical or β-sheet features (panels c and d). In the downward (I/II) cross peak (Figure 4a) region, we see two negative (bleach) features, with pump frequencies of 1620 and 1658 cm^−1^. These two frequencies correspond to amide I β-sheet and helical conformation, respectively. The corresponding probe frequencies (1540 and 1556 cm^−1^) then indicate which amide II frequencies are coupled with these two structures. Similarly, in the upwards (II/I) cross-peak region (Figure 4b), we see two corresponding bleach features at around (1530, 1630) and (1550, 1660) cm^−1^, however the frequencies do not correspond exactly. As discussed above, peaks in 2DIR spectra always present as a +/− pair dispersed on the probe axis. If the anharmonicity is small compared to the width of the peaks, the positive and negative features will overlap, which has the result of pushing the minimum of the bleach signal to a higher frequency than expected (see Figure 1). Additionally, excited state absorption of higher lying features can shift bleach minima back to lower frequencies. However, this complication is most significant on the probe axis, which means that in general the pump frequencies of minima are truthful reporters of peak positions. Thus, by taking a combination of the upwards and downwards cross-peak regions, we can accurately assign the frequencies of these two bleach features, with minimal intrusion from excited state absorption.

To aid in assigning these two bleach features, Figure 4c,d show integrations along the pump axis that intersect the bleach minima on the probe axis. The combined positions of the minima of these integrated cuts thus report the true positions of the amide I/amide II cross-peaks. The first peak intersects the amide I region at 1658 cm^−1^, and the amide II region at 1552 cm^−1^. Based on the amide I frequency [5,6], we are prompted to assign the peak at 1552 cm^−1^ to α-helix or other helical absorption.

However, a recent paper [31] suggested that silk I materials contain helical conformations consisting of repeated beta-turns. This structure is believed to be stabilized by intra-molecular hydrogen-bonds between the oxygen atom of the (i)-th glycine residue and the amide hydrogen atom of the (i + 3)-th alanine residue. From our results, we cannot determine whether the helical structure indicated by the double cross peak is due to an α-helix or a helical repeated beta-turn conformation, so we assign the 1552 cm^−1^ peak to the amide II of a generic helical structure absorbing at around 1658 cm^−1^, and hope our results will stimulate further experiment to settle the precise nature of the helical structure.

The same process finds an amide I frequency of 1620 cm^−1^ for the other peak, which gives the clear assignment of β-sheet absorption at 1530 cm^−1^. Note that the positions of the peaks differ slightly from those found in the diagonal cut, this is because excited state absorptions distorts the peak positions on both the diagonal cut and the cross peaks [32].

Although we observed in the diagonal spectrum (Figure 3a) a very strong, sharp feature at the pump frequency of 1517 cm^−1^, in both the upward and downwards amide I/II cross peak spectra there is a clear absence of a major feature pumping or probing at 1517 cm^−1^. If the 1517 cm^−1^ peak were an amide II vibration we would expect it to couple strongly with the amide I vibrations (as with the other features in this region), however the only evidence of any cross peak are three very weak features at (1610, 1517), (1640, 1517) and (1675, 1517) cm^−1^. Rather than an amide II vibration, we believe this feature is a tyrosine sidechain mode. Tyrosine makes up 5% (molar) of amino acids in the silk sample, and is known to have a strong absorbance around 1517 cm^−1^, due to a group vibration of the aromatic ring, for both for both natural and partially deuterated amino acids, with an extinction coefficient of ~400 M^−1^ cm^−1^, which similar to that of the amide I mode [5,6]. The integrated area of the 1517 cm^−1^ peak on the diagonal cut comes out as ~4% of the total amide I area, which agrees very well with the expected number of tyrosine oscillators. The feature is so prominent because all the oscillator strength is contained in a narrow (~10 cm^−1^ FWHM) peak, and presumably further enhanced by the µ^4^ scaling of 2DIR. Alongside the intense 1517 cm^−1^ vibration, Tyr has two weaker stretches in the region of 1595 to 1620 cm^−1^ [5,6]. The weak cross peak at (1610, 1517) cm^−1^ is thus likely to be a cross peak with one or more weaker Tyr modes. The cross peak at (1640, 1517) cm^−1^ coincides with the location of the random coil amide I vibrations. Finally, the (1675, 1517) cm^−1^ cross peak may well correspond with turn structures. This suggests that Tyr residues are mostly located in unstructured regions of the protein, in agreement with a proposed ‘templating’ mechanism [30,33], and consistent with expectations that the hydroxytoluene side group is too large to fit easily in β-sheet crystals. We intend to investigate this feature in more detail in the future. Putting all our findings together, we arrive at the assignments of the peaks in the FTIR spectrum given in Table 1.

## 3. Discussion

Amide II spectra of *B. mori* silk have been reported frequently, but there is surprisingly little consistency in the assignments of these spectra. In part, this is due to the differences between each sample, such as whether it is a film or a fiber, or whether it has been exposed to methanol to induce crystallization. Additionally, differences in acquisition method, such as choice of crystal for ATR measurements, will also have an effect [34]. In part, however, the confusion likely stems from the difficulty of assigning such a congested spectral region. Table 2 summarizes the amide II assignments for *B. mori* silk from an inexhaustive selection of publications.

Though they vary considerably, previous assignments have a few key trends: the ‘amorphous’ (whether it is random coil or other helical structures) feature tends to be assigned a region in the region of 1530–1540 cm^−1^, while the β-sheets are generally in the 1510–1530 cm^−1^ region. Most works did not explicitly assign the tyrosine sidechain peak, though several did assign a β-sheet peak at this frequency (1517 cm^−1^). Finally, peaks with frequencies above 1540 cm^−1^ are rarely assigned.

Our assignment of the amorphous helical peak to 1552 cm^−1^ is higher in frequency than all the linear IR experiments. We assigned a peak at 1530 cm^−1^ to β-sheets, while many linear spectroscopy investigations assign this region to amorphous structures. Finally, thanks to an absence of cross-peaks we assign the peak at 1517 cm^−1^ to a tyrosine sidechain, rather than the β-sheet assignment that linear spectroscopy often opts for. Although 2DIR produces very different peak widths and heights compared to linear spectroscopy, the peak centers should not shift, so assignments between the two should be comparable.

Such a disagreement with established literature forces us to question how reliable our assignments are. In Figure 4 we can directly see the cross-peaks with amide I transitions, whose assignments are uncontroversial [5,6]. This gives us a considerable degree of confidence in the assignment of the amide II peaks, and it is hard to conceive of a means of reproducing these cross-peaks using any of the assignments from linear spectroscopy. The exception to this is the assignment of the 1517 cm^−1^ peak to a Tyr sidechain. This assignment comes from an absence of a signal, leading to a necessarily weaker argument than one based on a peak’s presence. However, the presence of cross-peaks at (1610, 1517), (1640, 1517) and (1675, 1517) cm^−1^ strongly suggests that this is not a β-sheet, as it has often been previously assigned.

Given the strength of the cross-peak analysis, we are confident in our amide II assignments. Although we reach different assignments, previous results should not be discarded since, in most cases, the assignments from linear spectroscopy were produced by varying parameters such as tension in fibers, [12,13,18,19] film hydration[2,17] or methanol addition [15,16,17], and correlating changes in the amide II spectrum to changes in the amide I region. Such correlations tend to be very convincing, however we must caution that we recently discovered that spectral changes in the amide I β-sheet region can sometimes be due to random coil structures, and not β-sheets at all [7].

Rather than discarding previous assignments, instead we must take our findings as evidence how different the amide II spectrum of silk can appear, depending on sample type (native, regenerated, recombinant) and form (solutions, films, fibers) or spectroscopic method. Further, without direct comparison to known amide I data from either cross-peaks or correlations, it is extremely challenging to produce a definitive assignment for a given sample.

## 4. Materials and Methods

### 4.1. Silk-Film Preparation

Films were prepared using the native silk feedstock (NSF) from the middle-posterior (MP) sections of silk glands from commercially reared *B. mori* silkworms (four-way poly-hybrid cross of two Japanese and two Chinese strains) in their 5th instar. Specifically, silkworms during the early stages of cocoon construction were sacrificed by decapitation, allowing the two silk glands and haemolymph to be ejected into a Petri dish. One gland was selected and transferred to a second Petri dish and immersed in type I (distilled and deionised) water. Using a pair of tweezers, the gland was divided around the mid-point and the anterior portion (containing more sericin) was discarded. A second cut was made where the (wider) middle section started and the (relatively narrow) posterior section was also discarded.

The thin membrane was peeled off the MP gland section, using fine tweezers under a stereomicroscope, and the viscous NSF (around 0.15 g, containing around 0.035 g of predominantly fibroin) was transferred to a 20 mm × 20 mm polystyrene weighing boat. Around 2 to 3 mL of type I water was added, the weighing boat was loosely covered with tissue paper and allowed to stand at ambient temperature. The NSF initially dissolved into the water, then a film formed as the water evaporated. The film was allowed to dry under ambient conditions for a few days, before being transferred to a vacuum oven (still in the weighing boat). Drying to constant weight was completed over several hours at 60 °C under vacuum. Then, the film was peeled off the weighing boat and transferred to a sealed plastic bag for storage until required.

It is known that some sericin is produced within the MP gland section, although the majority is produced from the middle of the gland onwards [35,36]. Nevertheless, previous work [37] suggested that samples prepared in this way contained mainly fibroin (>97% *w*/*w*) with a negligible amount of sericin (<3% *w*/*w*) [37].

### 4.2. FTIR and 2DIR Spectroscopy

The 2D-IR spectra were recorded using a 10 kHz commercial time-domain spectrometer (2DQuickIR from Phasetech, Madison WI, USA) [21,38]. In brief, the spectrometer splits 100 fs broadband (~1450–1800 cm^−1^, produced from a Light Conversion Pharos, Orpheus and Lyra system) mid IR light pulses into pump and probe beams, which can be mechanically delayed relative to one another. The spectrum of pump pulse is shown in the Appendix A. The pump pulses are split into pairs of time delayed pulses using an acousto-optic pulse shaper. The pump and probe beams are focused and combined at the sample in a non-collinear geometry. In all spectra shown here, the pump and probe polarizations were orthogonal to one another. The transmitted probe light is then grating dispersed onto a high speed MCT detector array (JackHammer from Phasetech, Madison WI, USA). 2D spectra are produced by stepping the time delay and relative phase between the two pump pulses, and Fourier transforming the measured probe light with respect to this time delay. To remove the influence of scattered light on the spectra, a 4-fold phase cycling scheme was used. To reduce the number of time points required, the experiments were carried out in a partially rotating frame. The pump energy was ~1 µJ per pulse pair, and the probe was significantly weaker. The sample film was sandwiched between two CaF_2_ windows with no spacer. FTIR spectra (Vertex 70v from Bruker, Billerica MA, USA) were recorded of the same samples immediately before the 2D-IR spectra were measured. Both 2DIR and FTIR machines were purged with dry nitrogen to prevent absorption by water lines.

## 5. Conclusions

In this paper, we showed how broadband 2DIR spectroscopy can be used to disentangle the broad amide II band in silkworm films in a label-free and not invasive manner. Because the 2DIR signal scales as μ^4^, we resolved the presence of distinguished sub-bands in the amide II broad band. Since amide I and amide II modes of the same structure are strongly coupled, the excitation of the amide I vibration strongly affects the vibration of the amide II and vice versa. This leads to the appearance of signature off-diagonal cross-peaks. As one amide I vibration will correspond to one amide II, we used the cross-peak dependence on excitation or probe frequency to disentangle amide II sub-bands and to assign to specific secondary structures.

We thus found that the amide II (N-H) absorbs at around 1552 and at 1530 cm^–1^ for helical and β-sheet structures, respectively. We also observed a peak at 1517 cm^−1^ which could not be easily assigned to an amide II mode, and instead we tentatively assigned it to a Tyr sidechain. These results stand in contrast with previous findings from FTIR spectroscopy alone, highlighting the power of multidimensional spectroscopy for untangling convoluted spectra, and suggesting the need for extreme caution when assigning silk amide II spectra without corroborating data.

## Figures and Tables

**Figure 1 molecules-27-06275-f001:**
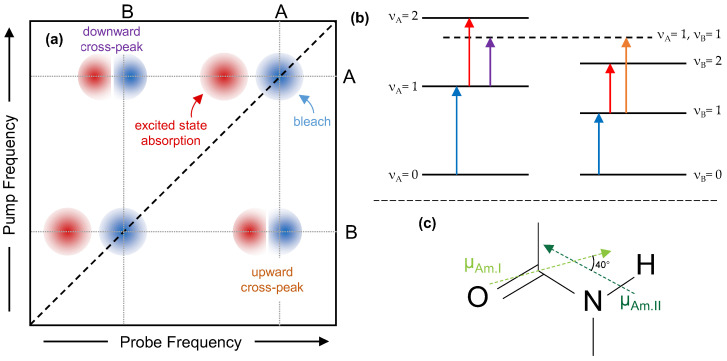
(**a**) Sketch of an ideal 2DIR spectrum of two coupled modes with frequencies A and B. Each mode produces a negative (blue) bleach feature on the diagonal (black dashed line), with a positive (red) excited state absorption at a lower frequency. Coupling between the two modes produces two set of cross-peaks, which also present as a bleach and excited state absorption doublet. Grey dotted lines show the connection between the diagonal features and the cross-peaks. Note that due to overlap with the excited state absorption, the minimum of the cross-peak bleach may not correspond exactly to the diagonal bleach on the probe axis, but on the pump axis, it will. (**b**) Schematic energy level diagram that produces the 2DIR spectrum shown in (**a**). (**c**) The sketch of the amide backbone structure, annotated with the transition dipole moments of the amide I and amide II vibrations.

**Figure 2 molecules-27-06275-f002:**
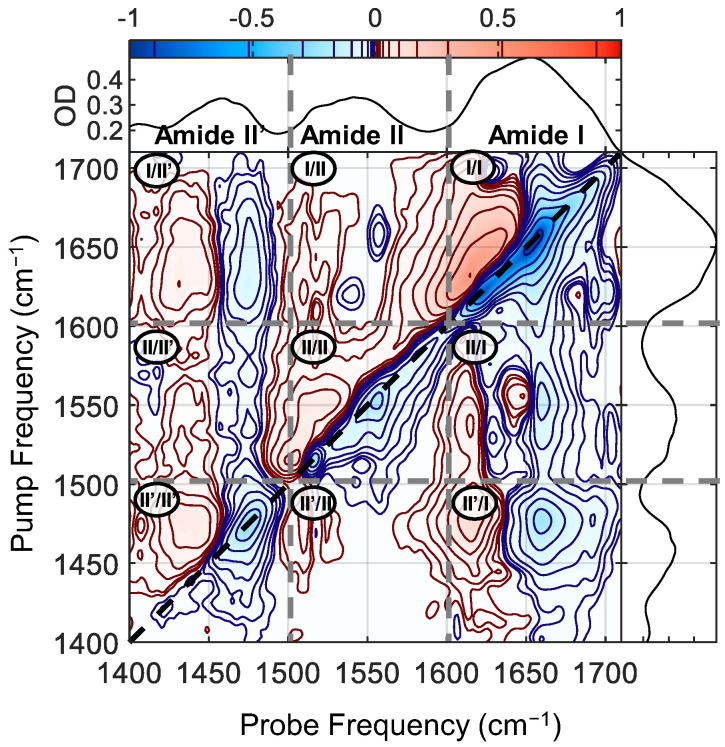
The broadband 2DIR spectrum of the silk film, after exposing it for 36 h to a heavy water saturated environment at RH of ~85%. The 2DIR spectrum can be divided in nine sectors: three regions where the same amide I/II/II’ vibrations are pumped and probed (diagonal regions), and six off-diagonal regions (cross-peak regions) that show the vibrational response of one amide region as a function of excitation at another. The labels in each sector are in the format of (pump/probe). The 2DIR spectrum is presented with a linear color scale (shown atop), but to enhance weak features the contours are spaced non-linearly (marked within the color bar). Intensity is normalized to the minimum of the diagonal bleach feature. The FTIR spectra are shown for reference both above and to the right of the 2D spectrum.

**Figure 3 molecules-27-06275-f003:**
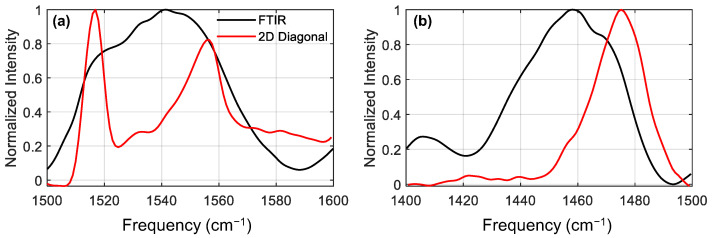
Comparison of the FTIR spectrum and the diagonal bleach slice of the 2DIR spectrum (multiplied by −1) for (**a**) the amide II region and (**b**) the amide II’ region. Diagonals from the 2DIR spectra are much sharper, thanks to their µ^4^ scaling as opposed to linear FTIR’s µ^2^ scaling.

**Figure 4 molecules-27-06275-f004:**
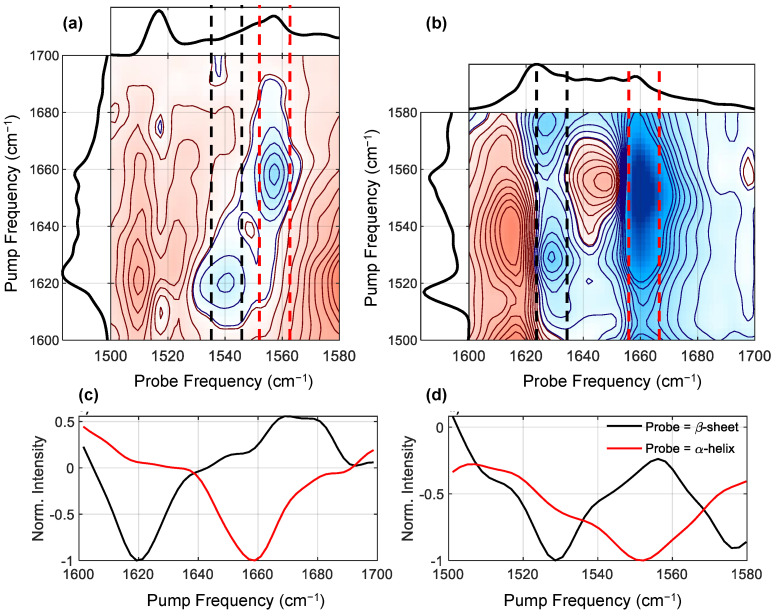
(**a**,**b**) Zoom of the amide I/II downward (**a**) and II/I upwards (**b**) cross peak regions. Compared to Figure 2, the color and contour scale is enhanced 10×, so that weak features are more easily discerned. The diagonal cuts are shown alongside both 2DIR spectra. (**c**,**d**) Normalized integrations along the pump axis of the cross-peak spectra intersecting either the helical feature (red), or β-sheet feature (black). The locations of the integrated regions are annotated on the 2DIR spectra in (**a**,**b**) by dashed lines.

**Table 1 molecules-27-06275-t001:** Assignments of *B. mori* silk from 2DIR.

Frequency (cm^−1^)	Assignment
1517	Tyr side chains
1530	Amide II β-sheet
1552	Amide II helical structure
1620	Amide I β-sheet
~1640	Amide I random coil
1658	Amide I helical structure
1695	Amide I β-sheet

**Table 2 molecules-27-06275-t002:** A selection of assignments for the amide II region for *B. mori* silk from 1D spectroscopy, alongside our findings. All ‘non β-sheet’ assignments are collected under the ‘amorphous’ label. ATR = attenuated total internal reflection FTIR spectroscopy, PM-IRLD = polarization modulated infrared linear dichroism. All frequencies are in cm^−1^.

Reference	Amorphous (Random Coil, Helical, etc.)	β-Sheet	Tyr	Method
This work	1552	1530	1517	2DIR, deuterated film
[2]	1534	1510, 1518, 1562	–	ATR, deuterated fibers
[12]	1550	1520	1515	PM-IRLD, film
[13]	1529	1510	1515	ATR, fiber
[14]	1535	1517	–	ATR, electrospun fiber
[15]	1539	1515	–	ATR, film
[16]	1540	1533	–	FTIR, film
[17]	1538	1528	–	ATR, film
[18]	1536	1512, 1551	–	FTIR, film
[19]	1547	1516	–	FTIR, fiber

## Data Availability

Data are available upon requests.

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
