# Peer review of "Broadband Multidimensional Spectroscopy Identifies the Amide II Vibrations in Silkworm Films"

_molecules, 2022, doi:10.3390/molecules27196275_

Round 1
Reviewer 1 Report
The authors used two-dimensional infrared spectroscopy to disentangle the broad infrared band in the amide II vibrational regions of Bombyx mori native silk films. According to the results obtained from 2D infrared spectra, they assigned the single amide II modes and correlated them to specific secondary structure. Although the infrared spectra of silk fibroin have been extensively studies, this m/s still gives the audience some novel information. However, there are several points should be clarified before it can be accepted for publication.
1. p.9, “the viscous NSF (around 0.15 g, containing around 0.035 g of predominantly fibroin)”. What is the exactly samples? Is it pure silk fibroin? I did not see any procedure to remove sericin (actually it is very difficult to distinguish sericin layer in silk gland and remove it from the silk fibroin). If the final film samples contain both silk fibroin and sericin, the total results and discussion will be meaningless.
2. The authors would be better explain the difference between current method and 2D infrared correlation spectroscopy.
3. As a basic knowledge, there is no alpha-helix in B. mori silk fibroin. Silk I in B. mori silk fibroin is a helical conformation, but no an alpha-helix.
4. During the measurement, there is still water in the sample. We know that even trace of water will affect the amide I and II bands a lot, especially when determining the exact location of the adsorption peak. Did the authors consider this?
Reviewer 2 Report
The article is well written addressing an important topic. The authors used two-dimensional infrared spectroscopy to disentangle the broad infrared band 12 in the amide II vibrational regions of Bombyx mori native silk films, identifying the single amide II 13 modes and correlating them to specific secondary structures. The methods have been detailed as needed, results are novel and conclusions have been found supported by evidence. The article is publishable in its current form.
Reviewer 3 Report
The authors have carefully carried out the experiments and presented the results neatly.
I have the following comments for further improvement of the paper.
I brief outline of the difference between the FTIR and 2D IR should be introduced in the introduction.
Please give a schematic for the transition from VA0 to VA1, and explain in the detail the mechanism of amide splitting.
What is the difference between µ4 and µ2 scaling? Please clearly mention it in the manuscript.
Author Response
Please see attach.
